# HIGH DIMENSIONAL CAUSAL INFERENCE WITH VARIATIONAL BACKDOOR ADJUSTMENT

## ABSTRACT

Backdoor adjustment is a technique in causal inference for estimating interventional quantities from purely observational data. For example, in medical settings, backdoor adjustment can be used to control for confounding and estimate the effectiveness of a treatment. However, high dimensional treatments and confounders pose a series of potential pitfalls: tractability, identifiability, optimization. In this work, we take a generative modeling approach to backdoor adjustment for high dimensional treatments and confounders. We cast backdoor adjustment as an optimization problem in variational inference without reliance on proxy variables and hidden confounders. Empirically, our method is able to estimate interventional likelihood in a variety of high dimensional settings, including semi-synthetic X-ray medical data. To the best of our knowledge, this is the first application of backdoor adjustment in which all the relevant variables are high dimensional.

## 1 INTRODUCTION

Understanding causal relationships is central to many scientific disciplines such as healthcare and economics. In these settings, professionals need to assess the importance of interventions (e.g. treatments, policy changes) on societal outcomes (e.g. patient health, economic well-being). However, determining interventional effects is challenging due to the presence of confounders, such as age, gender, and income of participants. One approach is to directly collect interventional data using randomized control trials (RCTs), but RCTs must be designed carefully (Schulz & Grimes, 2002) and are costly with respect to time and money (Sørensen et al., 2006). As an alternative, causal inference can be used to estimate interventional quantities from observational data alone (Pearl, 1995). Given modeling assumptions encoded in a DAG, Pearl's backdoor adjustment (Pearl, 2009b) estimates interventional likelihood by re-weighting the observational likelihood of the outcomes to block the influence of confounding variables.

This paper makes progress towards the application of backdoor adjustments for high-dimensional datasets. In many real-world scenarios, we frequently encounter treatments, outcomes, and confounders as high dimensional objects. For example, it is often the case that treatments are text embeddings of recommended procedures, outcomes are images such as X-ray and MRI screenings, and confounders are high dimensional genetic and environmental factors. High dimensional backdoor adjustment suffers from two major challenges: *intractability* in integrating out high-dimensional confounders and *expressivity* in learning non-linear dependencies from the observational distributions.

Prior work in this space can partly address these challenges. Inverse propensity weighting methods (Austin, 2011) can be used for high dimensional confounders, but do not generalize well to high dimensional treatments. Another common approach is using variational autoencoders (VAEs) (Kingma & Welling, 2013) as models of proxy confounding, where CEVAE is a notable example (Louizos et al., 2017). These models are overly expressive, in a sense, because they model unobserved confounders that are unidentifiable (Rissanen & Marttinen, 2021).

We propose Variational Backdoor Adjustment (VBA), a novel variational approach for backdoor adjustment under direct confounding. Our method uses variational inference to optimize a lower bound of the interventional likelihood and compute backdoor adjustment in high dimensions. We encounter and overcome the following key challenges:

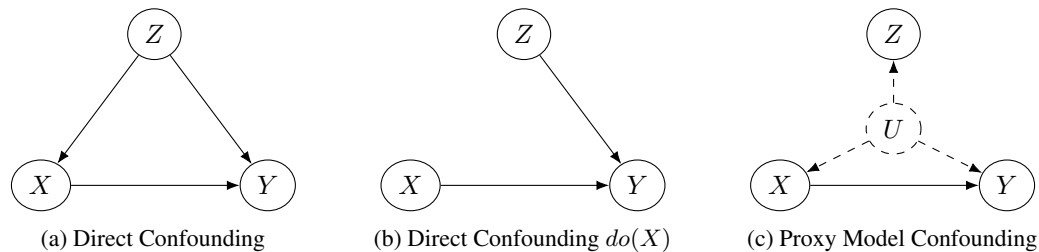

Figure 1: A visualization of the various discussed DAGs. Figure (a) and (c) show different approaches to modeling confounding. In this work, we opt for (a) due to identifiability. Figure (b) shows the $do$ operation graphically.

1. For high dimensional confounders and treatments, backdoor adjustment is either intractable to compute exactly or impractical to estimate with sampling due to high variance. We apply variational inference to efficiently obtain a more accurate approximate estimate.

2. Variational inference is typically applied in a latent variable setting in which the confounder in question is unobserved. Because the latent variable is unidentifiable, applying backdoor adjustment leads to an inconsistent estimate. We apply variational inference such that the latent space is restricted to the observed confounder.

3. However, in the new regime we cannot jointly optimize over all parameters, as is typical in VAEs. We introduce an optimization method that respects a latent space constrained by observation.

In our framework, we define three distributional components: encoder, decoder, and prior. Note that although we utilize standard VAE terminology to invoke similar components, our method is not to be confused with VAE due to different assumptions. These components must first be optimized separately to ensure identifiability, such that backdoor adjustment is correctly computed over the observed distributions. Once the components are optimized separately, the encoder can be further optimized to obtain better interventional density estimates.

This framework proves empirically effective at computing backdoor adjustment in a variety of synthetic and semi-synthetic settings. We construct a high dimensional linear Gaussian training set to empirically verify interventional density estimation with VBA. In this setting, we show that our optimization technique provides significant improvement over more naive estimation strategies. We then construct image datasets to test backdoor adjustment in more nonlinear settings. As a proof of concept, we demonstrate a potential application of VBA on X-ray medical data. We once again show in this setting the importance of the optimization technique used in VBA.

## 2 RELATED WORK

The existing body of literature pertaining to treatment effects is extensive (Shalit et al., 2017; Shi et al., 2019; Chen et al., 2019; Tesei et al., 2023). While these methods relate to individual treatment effects, backdoor adjustment is applied at the population level. Recent work uses neural mean embeddings (Xu & Gretton, 2022) for backdoor adjustment. While related, it is not applied to high dimensional outcomes. We shall also address related works that primarily fall into two approaches.

**Propensity Score Methods** In the literature, a common way to handle high dimensional confounding is by using propensity score methods (Austin, 2011). The key idea is that if a high dimensional confounder $z$ is sufficient for backdoor adjustment, then a single dimensional propensity score $g(z) = p(X = 1 \mid z)$ will suffice for backdoor adjustment (Rosenbaum & Rubin, 1983). Note that $X$ in this case is assumed to be a binary treatment. To our knowledge, there is no work applying propensity score methods to high dimensional treatments. While there may be no reason in principle that propensity score methods cannot be applied to such scenarios, they do not address the fundamental inference problem posed when $X$ is high dimensional; namely, that the conditional density $p(y \mid x, g(z))$ will have high variance due to a small effective sample size for each treatment. Shifting the burden of adjustment onto $g$ will not solve this issue for high dimensional treatments. Hence, we argue for the use of variational inference.

**Proxy Model VAE and Identifiability**   Past work applies variational autoencoders to causal inference, but with different modeling assumptions. Methods such as CEVAE (Louizos et al., 2017), VSR (Zou et al., 2020), and TEDVAE (Zhang et al., 2021) operate under the assumption that the true confounder is unobserved and only a proxy of the latent variable is observed, as depicted in Figure 1c. Recent work by Rissanen & Marttinen (2021) demonstrates that these techniques do not yield consistent estimates of causal effect because in general they cannot, in practice or even in theory, model unobserved latent variables correctly. Vanilla VAEs are unidentifiable (Locatello et al., 2019), which means that there exists infinite number of transformations on the latent variable that would emit the same marginal distribution. For this reason, performing backdoor adjustment over an unidentifiable latent variable can lead to completely inaccurate results. Recent works such as iVAE (Khemakhem et al., 2020) show that a factorized prior conditioned on additional observations give identifiability up to a class of transformations, and Intact-VAE (Wu & Fukumizu, 2021) exploits these results. In practice, the assumptions needed for identifiability cannot be verified for unobserved variables in real-world data. Rather than modeling a proxy of an unobserved confounder, we perform causal inference over observed variables only, as seen in Figure 1a. While the proxy model approach must argue that unobserved variables are identifiable, our approach is identifiable by definition because all variables are observed.

## 3   APPROACH

### 3.1   PRELIMINARIES

We are given variables $X, Y, Z$ as depicted in Figure 1a. We shall refer to $X, Y, Z$ respectively as treatment, outcome, and confounder. Suppose we are given probability density functions $p(z) = p(Z = z)$ and $p(y \mid x, z) = p(Y = y \mid X = x, Z = z)$. We can apply Pearl's backdoor adjustment (Pearl, 2009b) to obtain the interventional density

$$p(y \mid do(x)) = \sum_z p(z)p(y \mid x, z). \tag{1}$$

In practice, we will not know the prior $p(z)$ and outcome likelihood $p(y \mid x, z)$ but will assume access to an observational dataset $D = (x_i, y_i, z_i)_{i=1}^n$ consisting of $n$ triplets for $(x, y, z)$. The $do$ operator represents an intervention on the value of $X$, which severs the influence of $Z$ on $X$. The result is the distribution induced by modifying the DAG as seen in Figure 1b. Observe that the right-hand-side of the equation does not contain the $do$ operator, thus allowing interventional quantities to be obtained from observational data alone. Note that backdoor adjustment applies more broadly than to the DAG in Figure 1. In general, backdoor adjustment can be applied if the variables in question satisfy the *backdoor criterion* (Pearl, 2009a), which slightly broadens the scope of this work, as many DAGs can be equivalent to Figure 1a with respect to backdoor adjustment.

This formula alone is satisfactory in low dimensional settings when the number of confounders is limited. In high dimensional settings however, it is known that exact inference is intractable and the sum will be exponential in the dimension of $Z$ (Wang & Kwiatkowska, 2022). Naturally, for higher dimensions one may learn a conditional distribution $p(y \mid x, z)$ and generative model of the confounder $p(z)$. The interventional density $p(y \mid do(x))$ would then be approximated by drawing samples from the confounder $z \sim p(z)$ and computing an expectation over $p(y \mid x, z)$. However, it is well-known that a naive Monte Carlo estimate will have high variance in large dimensions because a sampled high dimensional $Z$ will almost never give high probability to $p(y \mid x, z)$ for a chosen $Y$ and $X$. Thus, even with a perfectly learned $p(z)$ and $p(y \mid x, z)$, inference of $p(y \mid do(x))$ using naive sampling is impractical.

### 3.2   VARIATIONAL BACKDOOR ADJUSTMENT

Variational inference is a commonly used technique for generative modeling tasks (Kingma & Welling, 2013; Rezende & Mohamed, 2015). In variational inference, the key idea is to approximate the posterior $p(z \mid x)$ with a simpler distribution, thus allowing marginal likelihood to be approximated with an evidence lower bound (ELBO). With variational inference, we can estimate a lower bound on

the interventional density given in Equation 1. Given some auxiliary "simple" distribution $q$, we have

$$\log p(y \mid do(x)) = \log \sum_z p(z)p(y \mid x, z) \tag{2}$$

$$= \log \mathbb{E}_{q(z \mid x, y)} \left( \frac{p(z)p(y \mid x, z)}{q(z \mid x, y)} \right) \tag{3}$$

$$\geq \mathbb{E}_{q(z \mid x, y)} \left( \log p(z) + \log p(y \mid x, z) - \log q(z \mid x, y) \right). \tag{4}$$

The inequality in Equation 4 is given by Jensen's inequality. In theory it is possible to condition $q$ on exclusively $x$ or $y$, but $q$ will be learned so it is best to parameterize it in the most expressive way. Intuitively, the purpose of $q(z \mid x, y)$ is as an encoder, giving high probability samples of $Z$, which will help decrease variance in high dimensions. Contrast this with the aforementioned Monte Carlo estimate in which we sampled from $p(z)$, leading to high variance. The penalty incurred by the encoder will be its KL divergence with the true prior distribution $p(z)$. It is well known in probabilistic inference that sampling is unbiased but has high variance, while variational inference will have some bias in exchange for much lower variance (Lange et al., 2022). Note that while our method utilizes variational inference, it is not a standard VAE because in Equation 4 the interpretation of $z$ is not as a latent variable. Instead, we have observed data on $Z$.

## 3.3 OPTIMIZATION METHOD

Because $z$ is not a latent variable, we cannot optimize the lower bound in the same manner as a VAE. To illustrate this, we shall first introduce our two step optimization method to perform Variational Backdoor Adjustment (VBA). After introducing our method and giving notation, it will become more clear why we cannot apply standard VAE "joint" optimization in our setting.

**Separate Training Phase** Let $p(z)$ and $p(y \mid x, z)$ be parameterized with a model of the prior $p_\theta(z)$ and decoder model $p_\gamma(y \mid x, z)$. We optimize these models with maximum likelihood objectives. Recall that the observational data is given by $D = (x_i, y_i, z_i)_{i=1}^n$. Each loss respectively will be computed over the dataset as $\mathbb{E}_D(\mathcal{L}(*))$.

$$\mathcal{L}_\theta^{\text{MLE}}(z) = -\log p_\theta(z) \tag{5}$$

$$\mathcal{L}_\gamma^{\text{MLE}}(x, y, z) = -\log p_\gamma(y \mid x, z) \tag{6}$$

Let $q_\phi(z \mid x, y)$ be an encoder model. It can also be optimized with maximum likelihood training, although it need not be limited in this way.

$$\mathcal{L}_\phi^{\text{MLE}}(x, y, z) = -\log q_\phi(z \mid x, y) \tag{7}$$

We can optimize these components separately using a gradient-based optimizer and obtain a lower bound on the interventional likelihood by plugging in the components as suggested by Equation 4. The model optimized by such loss functions is

$$f_{\phi,\theta,\gamma}(x, y) = \mathbb{E}_{q_\phi(z \mid x, y)} \left( \log p_\theta(z) + \log p_\gamma(y \mid x, z) - \log q_\phi(z \mid x, y) \right) \tag{8}$$

This step of VBA will be referred to as *separate training*.

**Finetuning Phase** Separate optimization will not lead to the tightest lower bound on interventional likelihood. We can further optimize the encoder to obtain more accurate interventional density estimation. Let $\hat{\theta}$ and $\hat{\gamma}$ be parameters optimizing their own respective MLE objectives in separate training. We give the following finetuning objective, which can be optimized with respect to $x, y \sim D$

$$\mathcal{L}_\phi^{\text{ELBO}}(x, y) = -\sum_{z_1', \dots, z_n' \sim q_\phi(z \mid x, y)} \left( \mathcal{L}_{\hat{\theta}}^{\text{MLE}}(z_j') + \mathcal{L}_{\hat{\gamma}}^{\text{MLE}}(x, y, z_j') - \mathcal{L}_\phi^{\text{MLE}}(x, y, z_j') \right) \tag{9}$$

Following separate training, we now optimize $\mathcal{L}^{\text{ELBO}}$ with gradient descent. We refer to this step as *finetuning* the encoder. This loss will optimize the expectation seen in Equation 4. Observe that the loss function differs from the typical ELBO objective seen in VAEs because the parameters $\theta, \gamma$ of the "decoder" $p_\gamma(y \mid x, z)$ and "prior" $p_\theta(z)$ are held fixed. We show experimentally that the additional finetuning phase improves finite sample performance compared to MLE separate training alone. The mechanism for this improvement is that the encoder is able to increase likelihood of the decoder $p(y \mid x, z)$ or prior $p(z)$ by generating less or more likely $z'$, as determined by the ELBO loss.

**Pitfall of Joint Optimization** Instead of fixing $\theta$ and $\gamma$ to their maximum likelihood estimates as shown in Equation 9, it may seem tempting to optimize all of the parameters "jointly" as follows

$$\mathcal{L}_{\phi,\theta,\gamma}^{\text{ELBO}}(x,y) = -\sum_{z_1',\ldots,z_n' \sim q_\phi(z|x,y)} \left( \mathcal{L}_\theta^{\text{MLE}}(z_j') + \mathcal{L}_\gamma^{\text{MLE}}(x,y,z_j') - \mathcal{L}_\phi^{\text{MLE}}(x,y,z_j') \right) \quad (10)$$

As previously explained, for VAEs the interpretation of $z$ in Equation 4 is as a latent variable. However, in our case, $z$ represents observed data, therefore backdoor adjustment necessitates that those quantities must be maximum likelihood estimates. Equation 10 is not guaranteed to converge to $\log p(y \mid do(x))$.

**Proposition 1** *There exists a Structural Causal Model (SCM) over $X, Y, Z$ such that for the optimal parameters $\phi^*, \theta^*, \gamma^* = \arg\min \mathcal{L}_{\phi,\theta,\gamma}^{ELBO}$, we have $f_{\phi^*,\theta^*,\gamma^*}(x,y) = \log p(y \mid x)$.*

Proof of this proposition is in Appendix B. Intuitively, "joint" optimization does not work because the distribution over $Z$ will become untethered from the data, which means we are no longer performing backdoor adjustment. In the loss of Equation 10, because $\theta$ and $\gamma$ are not fixed, we can minimize the loss by making the KL divergence between $q_\phi(z \mid x, y)$ and $p_\theta(z)$ arbitrarily small while $p_\gamma(y \mid x, z)$ loses dependence on $Z$. This is very similar to a problem in VAEs called posterior collapse (Lucas et al., 2019), but in our situation it makes estimating backdoor adjustment impossible.

**Implementation Details** In this framework, each distribution can be parameterized as needed depending on the setting. In general, $p_\theta(z)$ will be a generative model to estimate the density of a high dimensional confounder. Depending on the circumstance, $p_\gamma(y \mid x, z)$ and $q_\phi(z \mid x, y)$ can be conditional generative models or vanilla neural networks. In most cases vanilla neural networks are sufficient, but in some cases $Z$ or $Y$ may contain significant exogenous stochasticity that will be better captured with conditional generative models.

## 4 EXPERIMENTS

Evaluation is one of the most challenging aspects to causal inference because the real-world does not grant access to ground truth causal effect. This is because we do not have a complete understanding of the data generating process, so if some outcome is observed, we do not have access to the counterfactual outcome given different circumstances. Some refer to this issue as the *fundamental problem of causal inference* (Holland, 1986).

To gain access to ground truth, one must make assumptions about the data generating process at the expense of realism. In this work, we attempt a balanced approach by evaluating at different points along this tradeoff. First, we evaluate our method on high dimensional linear Gaussian data. In this setting, ground truth causal effect can be solved for analytically. For a more realistic high dimensional setting that incorporates image data, we design a toy example involving MNIST images. Unfortunately, this setting does not have ground truth causal effect, but we can demonstrate qualitative improvements when using backdoor adjustment over naively modelling the observational distribution $p(y \mid x)$. Finally, we apply variational backdoor adjustment to a synthetic dataset containing real X-ray images and show the benefit of finetuning over separate training. In our empirical image experiments, we can only make qualitative observations about whether we estimate the true interventional density. Our main claim to the accuracy of VBA relies on experiments using linear Gaussian data. The purpose of these experiments can be summarized as follows:

- **Linear Gaussian**: shows that VBA is more accurate than sampling, and it consistently converges to the correct estimate.
- **MNIST**: gives a high dimensional image setting and compares VBA, which estimates the interventional distribution, to naively modelling the observational distribution.
- **X-ray**: presents a medical scenario and demonstrates the benefit of the "finetuning" mechanism over "separate training" alone.

### 4.1 LINEAR GAUSSIAN

To evaluate the VBA density estimation against ground truth, we consider the following setting. Suppose the relationships between variables are linear and the base distributions are Gaussian. The

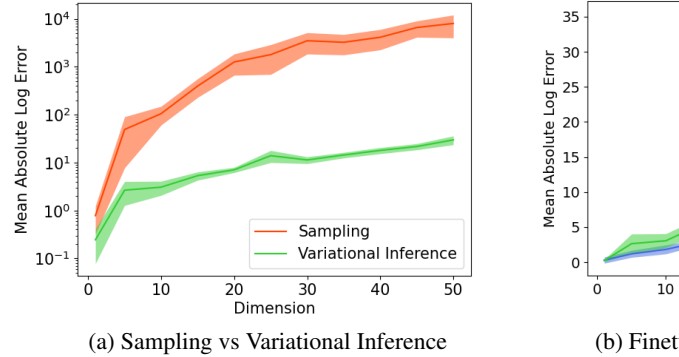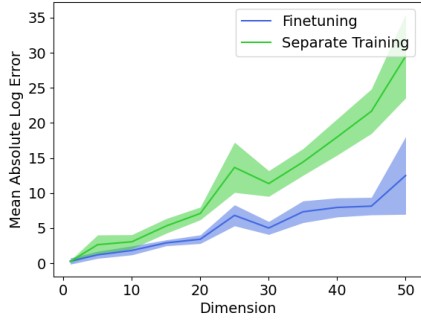

(a) Sampling vs Variational Inference
Log Error Comparison

(b) Finetuning vs Separate Training
Log Error Comparison

Figure 2: Figures (a) and (b) contrast variational inference with naive sampling at inference time. Variational inference proves beneficial, especially as dimensionality increases. Finetuning yields a better interventional density estimate than separate training. We average results over 10 random generations of data.

structural equations for $X, Y, Z$ will take the following form

$$Z := \mathcal{N}(0,1) \qquad X := c_1 Z + \mathcal{N}(0, \sigma_1) \qquad Y := c_2 X + c_3 Z + \mathcal{N}(0, \sigma_2) \qquad (11)$$

The linear Gaussian setting allows for fully analytic solutions to ground truth interventional density. The ground truth for interventional distribution is given by

$$y \mid do(x) \sim \mathcal{N}\left(c_2 x, \sqrt{c_3^2 + \sigma_2^2}\right) \qquad (12)$$

and we can compare it with the observational distribution

$$y \mid x \sim \mathcal{N}\left(\left(\frac{c_1 c_3}{(c_1 + \sigma_1^2)} + c_2\right) x, \sqrt{\frac{c_3^2 \sigma_1^2}{c_1^2 + \sigma_1^2} + \sigma_2^2}\right) \qquad (13)$$

Constants $c_1, c_2, c_3$ can be thought of as determining the strength of the causal relationships. It therefore follows that the mean of Equation 12 is only dependent on $c_2$ because $c_2$ determines the causal relationship between $X$ and $Y$. The mean of Equation 13 also contains $c_1$ and $c_3$, which are responsible for confounding. We give a concrete example of the difference between observational and interventional distribution in Figure 7 of Appendix A. So as not to draw conclusions based on cherry-picked parameters, for each experiment we randomly sample and fix $c_1, c_2, c_3$. We sample

Table 1: Linear Gaussian Results. We report values in nats and report the mean and standard error over 5 random seeds. The top row is mean absolute error from the ground truth for $\mathbb{E}_{x,y \sim D} \log p(y|do(x))$. Rows 2-4 below are decomposed from the ELBO. Row 5 gives the likelihood of the encoder. Finetuning the encoder $q$ gives better results as opposed to generating maximum likelihood $z$. Lower likelihood samples of $z$ results in a tighter ELBO.

| | In Distribution | | Out of Distribution | |
|---|---|---|---|---|
| | Separate Training | Finetuned | Separate Training | Finetuned |
| Ground Truth MAE ($\downarrow$) | 5.58 (0.003) | 1.89 (0.001) | 11.74 (0.005) | 7.51 (0.001) |
| $\mathbb{E}_{x,y \sim D} \mathbb{E}_{q(z\mid x,y)} \log p(z)$ | -21.17 (0.003) | -20.83 (0.000) | -129.41 (0.001) | -110.82 (0.000) |
| $\mathbb{E}_{x,y \sim D} \mathbb{E}_{q(z\mid x,y)} \log p(y \mid x, z)$ | -21.80 (0.001) | -22.14 (0.002) | -26.17 (0.003) | -34.13 (0.006) |
| $\mathbb{E}_{x,y \sim D} \mathbb{E}_{q(z\mid x,y)} \log q(z \mid x, y)$ | 17.20 (0.002) | 6.27 (0.001) | 17.20 (0.002) | 6.28 (0.001) |
| $\mathbb{E}_D[\log q(z \mid x, y)]$ | 14.97 | 4.17 | 3.28 | -25.02 |

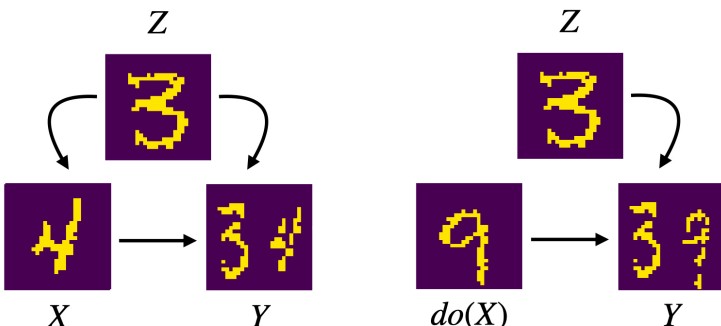

Figure 3: Data generating process for MNIST dataset. On the left is the observational distribution in which $Y$ can only have successive numbers. On the right, interventions on $X$ can produce images $Y$ with two numbers not seen on the left.

these constants in a manner that emphasizes the difference between interventional and observational distribution. Linear Gaussian hyperparameters are sampled as

$$c_1 \sim \mathcal{U}(0,5) \quad c_2 \sim \mathcal{U}(0,3) \quad c_3 \sim \mathcal{U}(-5,-10) \quad \sigma_1 \sim \mathcal{U}(0,1) \quad \sigma_2 \sim \mathcal{U}(0,3). \quad (14)$$

We obtain a high dimensional linear Gaussian dataset by repeatedly sampling a set of parameters for each dimension and treating each dimension independently. To perform backdoor adjustment on linear Gaussian data, we model the confounder with a VAE and both encoder and decoder with Gaussians parameterized by MLPs.

Figure 2a shows the difference between sampling and variational inference. For 10 unique generations of linear Gaussian data, we train the respective components using their own MLE objective (also known as separate training). At inference time, we compare two approaches for obtaining $\log p(y \mid do(x))$: sampling from the prior $z \sim p(z)$ and computing an average over $\log p(y \mid x, z)$, or sampling from the encoder $z \sim q(z \mid x, y)$ and utilizing variational inference given by Equation 4. Variational inference outperforms naive sampling as the dimensionality increases.

We use the same analysis to compare optimization methods. Figures 8b and 2b show a substantive increase in performance when finetuning the encoder, and the performance is robust in higher dimensions. To better understand why finetuning gives better estimates, we focus on a single run at a fixed dimension. We examine 15 dimensional linear Gaussian data generated with the process previously described, and Table 1 demonstrates that utilizing the finetuning objective for the encoder achieves log-likelihood estimates closer to the ground truth. We generate data out-of-distribution by intervening on $Z$ and setting it to a $\mathcal{U}(-7,7)$ variable. This in turn generates out-of-distribution $X$ and $Y$ by construction. These results analyze the components given by the formula for variational backdoor adjustment (Equation 4) and show the reason for increased performance: optimizing the encoder gives it the flexibility to give lower likelihood to samples without dramatically sacrificing likelihood of the decoder and prior. This optimization results in a tighter bound on Equation 4, and therefore a more accurate interventional likelihood.

## 4.2 MNIST

To demonstrate that our method can work on images, we construct a synthetic dataset of binary MNIST images that simulate confounding. The setup is as follows: sample a random image $Z$, sample an image $X$ such that $X$ is the consecutive number after $Z$, and concatenate them together to form image $Y$. The causal structure is depicted in Figure 3. In the training data, $Y$ will only contain concatenated images with successive numbers due to the confounding of image $Z$. However, by training with backdoor adjustment, the model can learn that $Y$ can consist of any two numbers given an intervention on $X$. For observational density $p(y \mid x)$, we train a conditional VAE with 50 latents and 2 hidden layers, each with 200 hidden units. For the interventional density $p(y \mid do(x))$, we apply VBA, parameterizing the encoder as a vanilla neural net, and the decoder and prior as VAEs with 50 latents and 2 layers of 200 hidden units. Because the data is binary, we sample in training using the Gumbel-Softmax trick (Jang et al., 2016), which allows gradient descent through discrete samples. The difference between observational and interventional density is born out in Figure 4. It

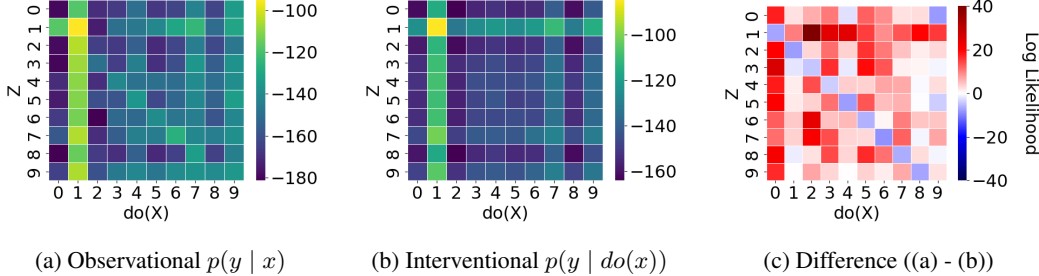

(a) Observational $p(y \mid x)$ (b) Interventional $p(y \mid do(x))$ (c) Difference ((a) - (b))

Figure 4: We generate interventional data for MNIST as seen on the right of Figure 3. Figures (a) and (b) show the respective log-likelihood, where the axes delineate how $Y$ is generated. For example, $Z = 3$ and $do(X = 9)$ will produce an image $Y$ reassembling a "39". The key result is (c), where we see that in general, interventional likelihood will give higher likelihoods. The blue diagonal in (c) shows that images with consecutive numbers have higher likelihood in the observational distribution.

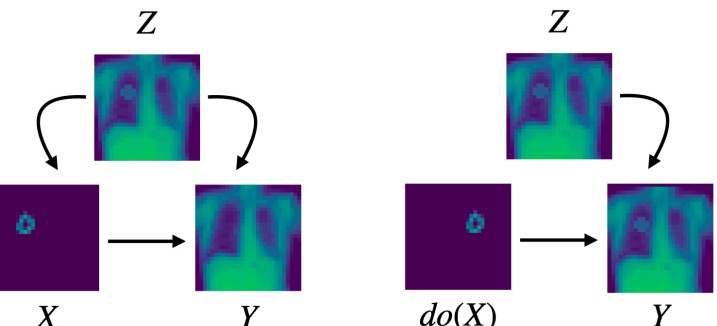

Figure 5: Data generating process for Causal X-ray dataset. The small circle in patient $Z$ is the tumor. On the right, intervening on the treatment will make recovery rate independent of whether the annotation is correctly placed, i.e. the accuracy of the doctor.

shows that using variational backdoor adjustment, images with non-consecutive numbers in $Y$ will be more likely under the interventional density than under a naively estimated observational density.

## 4.3 CAUSAL X-RAY

**Setup** We introduce a new high dimensional dataset containing X-ray images to emulate a potential real-world scenario. The data models the following situation: a patient receives an initial X-ray scan ($Z$), a doctor provides annotations indicating location for treatment ($X$), and the patient obtains a new X-ray scan following treatment ($Y$). The treatment is administered with a certain accuracy with respect to location based on the patient X-ray ($Z \rightarrow X$), and when administered, the treatment has a certain level of efficacy ($X \rightarrow Y$). The goal of backdoor adjustment is to isolate the effectiveness of the treatment and ignore the accuracy of how it is administered.

To generate the data, we utilize chest X-ray scans from MedMNIST (Yang et al., 2021), and apply a slight augmentation to add synthetic "tumors". A tumor is a circle composited with the image, and it occurs on the left or right lung with equal probability. The treatments are annotations that are synthetically created through shrinking MNIST zero digits to approximately the size of a tumor (see Figure 5 for visual). It is placed either on the left or right (either accurately or inaccurately depending on whether the tumor is placed on the left or right). In this setup, we assume that the doctor administers the treatment accurately 60% of the time. The outcome will be an X-ray image where if the treatment is effective and administered accurately, the patient will recover with 50% probability, resulting in an X-ray image without the tumor. If the treatment is administered incorrectly, the tumor will always remain in the patient.

**Results** We parameterize VBA for this task in the following manner. We model the confounder images $Z$ with an auto-regressive flow model called FFJORD (Grathwohl et al., 2018). We opt for this model

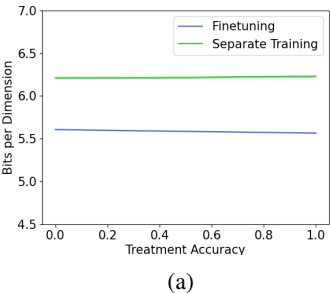 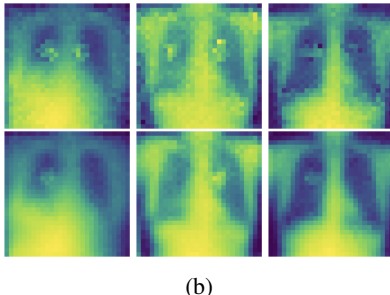

|           (a)           |           (b)           |

Figure 6: **Left:** Performance on Causal X-ray measured in bits/dim (lower is better). Estimates do not vary with treatment accuracy, as expected in the interventional density. **Right:** Samples drawn from separately trained encoder (top row) and finetuned encoder (bottom row). The images from the finetuned encoder are more smoother and more realistic, as they do not contain "double" tumor artifacts.

because it gives exact likelihood estimates in continuous settings and is amongst the state-of-the-art models for density estimation. Because in this setting $Z$ and $Y$ are closely related, parameterizing the encoder $p(z \mid x, y)$ and $p(y \mid x, z)$ with vanilla deep neural networks using a Gaussian predictive distribution with diagonal covariance is sufficient. To evaluate the efficacy of backdoor adjustment, we generate new interventional data in which we change the accuracy of the doctors treatments. In the training data, treatment occurs at the location of the tumor 60% of the time by construction. We can vary this rate when generating test data, and because treatment $X$ as no dependence on patient $Z$ in the interventional distribution, the likelihood should not significantly change.

Figure 6a shows little dependence between treatment accuracy and reported bits per dimension of VBA. It also shows that VBA with finetuning performs strictly better than separate training. We investigate this difference in performance by visualizing samples from each encoder. For a given $X, Y$, the finetuned encoder produces smoother, more realistic images $Z$. Better looking images along with greater likelihoods are indications of a better generative model and a more accurate estimate of $p(y \mid do(x))$. Intuitively, this is because a separately trained encoder must fit to the data, but the finetuned encoder is optimized to better fit with the prior $p(z)$ and decoder $p(y \mid x, z)$ to obtain a better lower bound. We confirm these intuitions in Table 2, which explicitly shows these trade-offs. Finetuning makes encoder samples $z$ more likely, but in turn also improves prior and decoder likelihood. Interestingly, the opposite effect is observed in Table 1, where lower likelihood samples $z$ improved the ELBO. We restate the importance and power of optimizing interventional density in this manner, since the interventional density estimate can be improved in multiple ways.

Table 2: Causal X-ray Results. We report in nats, with mean and standard error averaged over the test Causal X-ray data for 5 runs. The rows given are decomposed from the terms in the ELBO. Finetuning the encoder $q$ outperforms $z$ sampled from MLE estimates of $q$. In this case, higher likelihood samples of $z$ result in a tighter ELBO.

|  | Separate Training | Finetuned |
|---|---|---|
| $\mathbb{E}_{x,y\sim D}\mathbb{E}_{q(z\mid x,y)} \log p(z)$ | 979.67 (1.67) | 2012.51 (0.10) |
| $\mathbb{E}_{x,y\sim D}\mathbb{E}_{q(z\mid x,y)} \log p(y \mid x, z)$ | 1705.91 (0.23) | 1837.63 (0.08) |
| $\mathbb{E}_{x,y\sim D}\mathbb{E}_{-q(z\mid x,y)} \log q(z \mid x, y)$ | -1923.18 (0.08) | -2589.35 (0.08) |

## 5 CONCLUSION

We explore the various challenges that surface when trying to compute backdoor adjustment in high dimensions. We show that using variational inference, it is possible to compute the backdoor formula in the presence of high-dimensional treatments and confounders. While variational inference is typically used in a latent variable setting, we show with a novel optimization framework that variational inference can be used as a powerful tool to compute identifiable quantities in causal inference.

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
