## A  SUPPLEMENTAL LINEAR GAUSSIAN FIGURES

First we show here a useful visual to gain intuition about the linear Gaussian setting. We visualize the distributions given by Equation 11 using concrete values: $c_1 = 3, c_2 = 2, c_3 = -6, \sigma_1 = 0.5, \sigma_2 = 1$. We observe the potential difference between observational and interventional densities. In 7b and 7c, we depict two distributions that are consistent with the observed joint distribution depicted in 7a. Without the assumptions made by backdoor adjustment, we cannot estimate the interventional density seen in 7a.

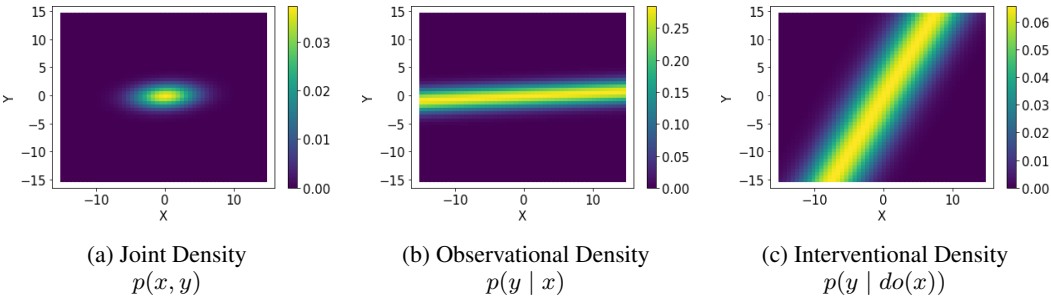

(a) Joint Density
$p(x, y)$

(b) Observational Density
$p(y \mid x)$

(c) Interventional Density
$p(y \mid do(x))$

Figure 7: Linear Gaussian Visualization

Below are figures that give a different view on Figures 2a and 2b. Instead of plotting the error we give the log likelihood plotted with the ground truth interventional likelihood. As given by the ELBO, our estimates will be a lower bound in expectation of the ground truth.

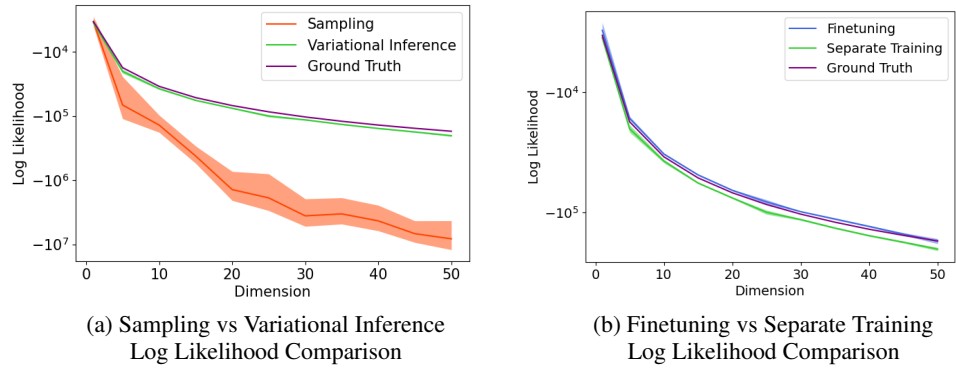

(a) Sampling vs Variational Inference
Log Likelihood Comparison

(b) Finetuning vs Separate Training
Log Likelihood Comparison

Figure 8: Linear Gaussian Results

## B  PROOF OF PROPOSITION 1

Consider the objective given in Equation 10.

$$\mathcal{L}_{\phi,\theta,\gamma}^{\text{ELBO}}(x, y) = - \sum_{z_1', \dots, z_n' \sim q_\phi(z|x,y)} \left( \mathcal{L}_\theta^{\text{MLE}}(z_j') + \mathcal{L}_\gamma^{\text{MLE}}(x, y, z_j') - \mathcal{L}_\phi^{\text{MLE}}(x, y, z_j') \right) \quad (15)$$

There exists parameters $\phi, \theta, \gamma$ such that

$$q_\phi(z \mid x, y) = p_\theta(z) \quad (16)$$

$$p_\gamma(y \mid x, z) = p(y \mid x) \quad (17)$$

Given these parameters, we get the following ELBO.

$$\mathbb{E}_{q_\phi(z|x,y)} \left( \log p_\theta(z) + \log p_\gamma(y \mid x, z) - \log q_\phi(z \mid x, y) \right) \quad (18)$$

$$= \mathbb{E}_{q_\phi(z|x,y)} \left( \log p(y \mid x) \right) \quad (19)$$

$$= \log p(y \mid x) \quad (20)$$

Thus, when $E_{p(x,y)} \log p(y \mid x) > E_{p(x,y)} \log p(y \mid do(x))$, the model $f_{\phi,\theta,\gamma}(x,y)$ will converge to $p(y \mid x)$. We can show by construction that this can occur for some SCM. Let $\mathcal{B}$ denote a Bernoulli distribution. Consider a simple example in which variables $X, Y, Z$ induce distributions

$$p(z) = \mathcal{B}(1/2) \tag{21}$$

$$p(x \mid z) = \begin{cases} \mathcal{B}(3/4) & \text{if } z, \\ \mathcal{B}(1/2) & \text{if } \neg z \end{cases} \tag{22}$$

$$p(y \mid x, z) = \begin{cases} \mathcal{B}(1/2) & \text{if } x \vee z, \\ \mathcal{B}(3/4) & \text{if } \neg x \wedge \neg z \end{cases} \tag{23}$$

It then follows from this example that

$$E_{p(x,y)} \log p(y \mid x) \tag{24}$$
$$= (1/8)\log(1/3) + (1/4)\log(2/3) + (5/16)\log(1/2) + (5/16)\log(1/2) \tag{25}$$
$$> (1/8)\log(3/8) + (1/4)\log(5/8) + (5/16)\log(1/2) + (5/16)\log(1/2) \tag{26}$$
$$= E_{p(x,y)} \log p(y \mid do(x)) \tag{27}$$

## C    PROOF OF PROPOSITION 2

In this proof we shall refer to the lower bound derived in Equation 4 as the Backdoor ELBO (BELBO).

$$\text{BELBO} = \mathbb{E}_{q(z|x,y)} \left( \log p(z) + \log p(y \mid x, z) - \log q(z \mid x, y) \right) \tag{28}$$

First we shall find the Jensen gap.

$$\log p(y \mid do(x)) = \log \sum_{z'} p(z')p(y \mid x, z') \tag{29}$$

$$= \mathbb{E}_{q(z|x,y)} \left( \log p(z) + \log p(y \mid x, z) - \log q(z \mid x, y) \right) \tag{30}$$

$$+ \mathbb{E}_{q(z|x,y)} \left( \log q(z \mid x, y) - \log \left( \frac{p(z)p(y \mid x, z)}{\sum_{z'} p(z')p(y \mid x, z')} \right) \right) \tag{31}$$

$$= \text{BELBO} + \text{KL} \left( q(z \mid x, y) \parallel \frac{p(z)p(y \mid x, z)}{\sum_{z'} p(z')p(y \mid x, z')} \right) \tag{32}$$

$$= \text{BELBO} + \text{KL} \left( q(z \mid x, y) \parallel p(z \mid do(x), y) \right) \tag{33}$$

Note that to arrive at $p(z \mid do(x), y)$ we utilize

$$p(z)p(y \mid x, z) = p(z \mid do(x))p(y \mid do(x), z) \tag{34}$$
$$= p(z, y \mid do(x)) \tag{35}$$

using do calculus. Rewriting we have

$$\text{BELBO} = \log p(y \mid do(x)) - \text{KL} \left( q(z \mid x, y) \parallel p(z \mid do(x), y) \right) \tag{36}$$

Finetuning loss $\mathcal{L}_{\phi}^{\text{ELBO}}$ maximizes the BELBO, which will obtain an optimum when the KL term is zero. Because we have a consistent estimate of $p(z)$ and $p(y \mid x, z)$, the KL term will approach zero if $q_{\phi}(z \mid x, y)$ is parameterized as a family of distributions that includes $p(z \mid do(x), y)$. An expressivity assumption on the encoder is made in VAE consistency arguments and is also necessary for our proof.

## D    PITFALL OF FULLY JOINT TRAINING

As state in Proposition 1 we cannot simply optimize the parameters of the prior $\theta$ and decoder $\gamma$, and that they must remain fixed in the objective $\mathcal{L}^{\text{ELBO}}$ given in Equation 9. If these parameters are freely optimized in a fully joint manner, the backdoor adjustment will not be computed correctly because $Z$ will not be constrained by observation. Below, we have a plot similar to Figure 8b.

This figure empirically shows that naively optimizing Equation 9 with fully joint training, that is optimizing all the parameters, will result in estimation of the wrong query. It can be seen that fully joint training is estimating $p(y \mid x)$ rather than $p(y \mid do(x))$

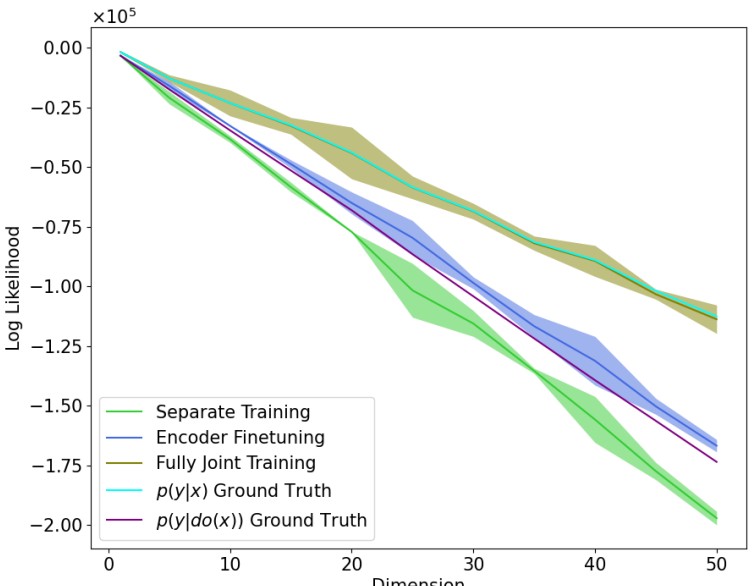

Figure 9: Pitfall of Fully Joint Training

## E    PROOF OF LINEAR GAUSSIAN ANALYTICAL SOLUTIONS

We prove the solutions found for the interventional and observational distributions seen in Equation 12 and Equation 13.

### E.1    INTERVENTIONAL

We aim to show that

$$y \mid do(x) \sim \mathcal{N}\left(c_2 x, \sqrt{c_3^2 + \sigma_2^2}\right)$$

First observe that the setup seen in Equation 11 entails $z \sim \mathcal{N}(0, 1)$ and $y \mid x, z \sim \mathcal{N}(c_2 x + c_3 z, \sigma_2)$. We can thus compute

$$p(y \mid do(x)) = \sum_z p(z) p(y \mid x, z) \tag{37}$$

$$= \sum_z \left( \frac{1}{\sqrt{2\pi}} e^{\frac{-z^2}{2}} \right) \left( \frac{1}{\sigma_2 \sqrt{2\pi}} e^{\frac{-(y - c_2 x - c_3 z)^2}{2\sigma_2^2}} \right) \tag{38}$$

$$= \frac{1}{\sqrt{(c_3^2 + \sigma_2^2) 2\pi}} e^{\frac{-(y - c_2 x)^2}{2(c_3^2 + \sigma_2^2)}} \tag{39}$$

The same proof can be found in Rissanen & Marttinen (2021) Appendix A Equation 12.

### E.2    OBSERVATIONAL

To compute $p(y \mid x)$ in this setting, we must express $Y := c_2 X + c_3 Z + \mathcal{N}(0, \sigma_2)$ in terms of only $X$. To do this, we must express $Z$ in terms of $X$. It is known that this swap can be performed using conjugate priors (Fink, 1997; Atkinson et al., 2022). We find

$$Z \mid X = x \sim \mathcal{N}\left( \frac{c_1 x}{c_1^2 + \sigma_1^2}, \sqrt{\frac{\sigma_1^2}{c_1^2 + \sigma_1^2}} \right)$$

Through simple addition and scaling properties of Gaussians, we combine the above with the definition of $Y$ to obtain

$$y \mid x \sim \mathcal{N}\left(\left(\frac{c_1 c_3}{(c_1 + \sigma_1^2)} + c_2\right)x, \sqrt{\frac{c_3^2 \sigma_1^2}{c_1^2 + \sigma_1^2} + \sigma_2^2}\right)$$

# F  IMAGE SAMPLES

In our generative models, we can draw samples to qualitatively ensure that the correct distribution is being learned.

## F.1  MNIST SAMPLES

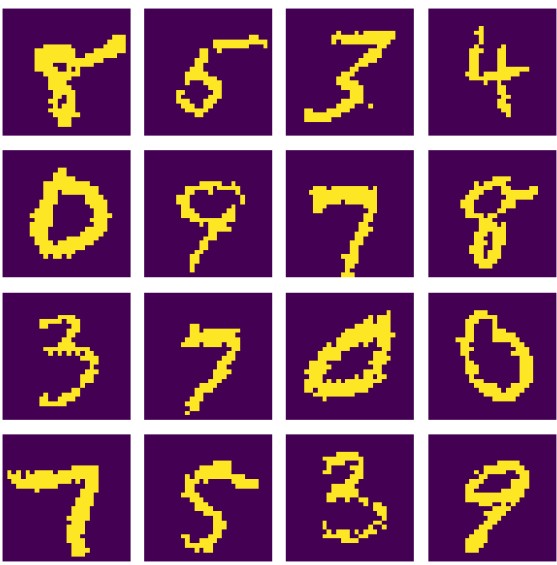

Figure 10: Examples of $X$ from MNIST dataset.

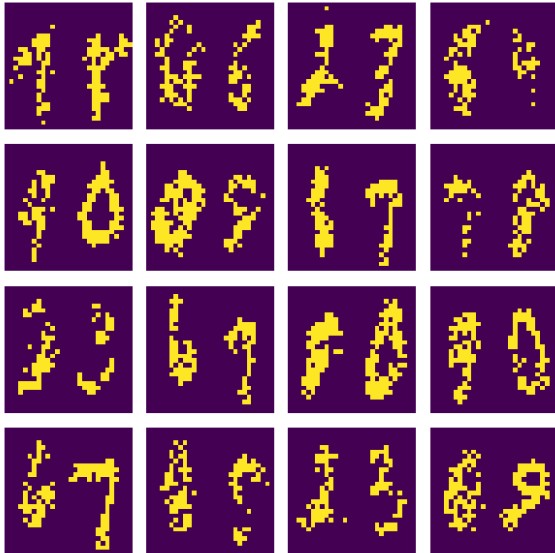

Figure 11: Samples from $p(y \mid x)$, with $X$ given by Figure 10. Here, we see that the numbers generated are successive because that is how $Y$ is observed.

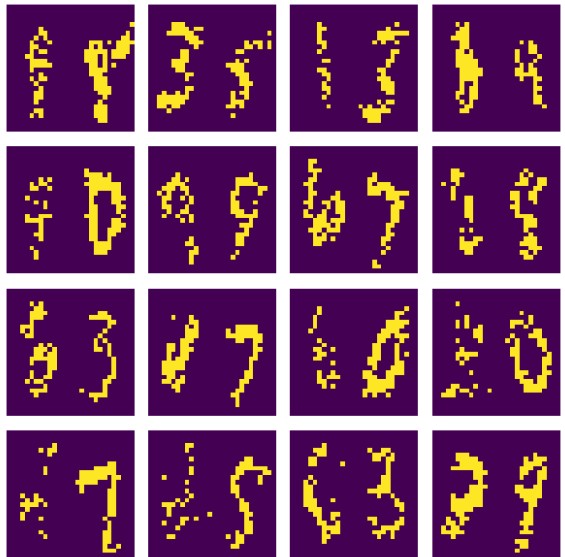

Figure 12: Samples from $p(y \mid do(x))$, with $X$ given by Figure 10. Here, we are able to sample $Y$ such that digits are not successive, which indicates the difference between the observational and interventional distributions.

## F.2 CAUSAL X-RAY SAMPLES

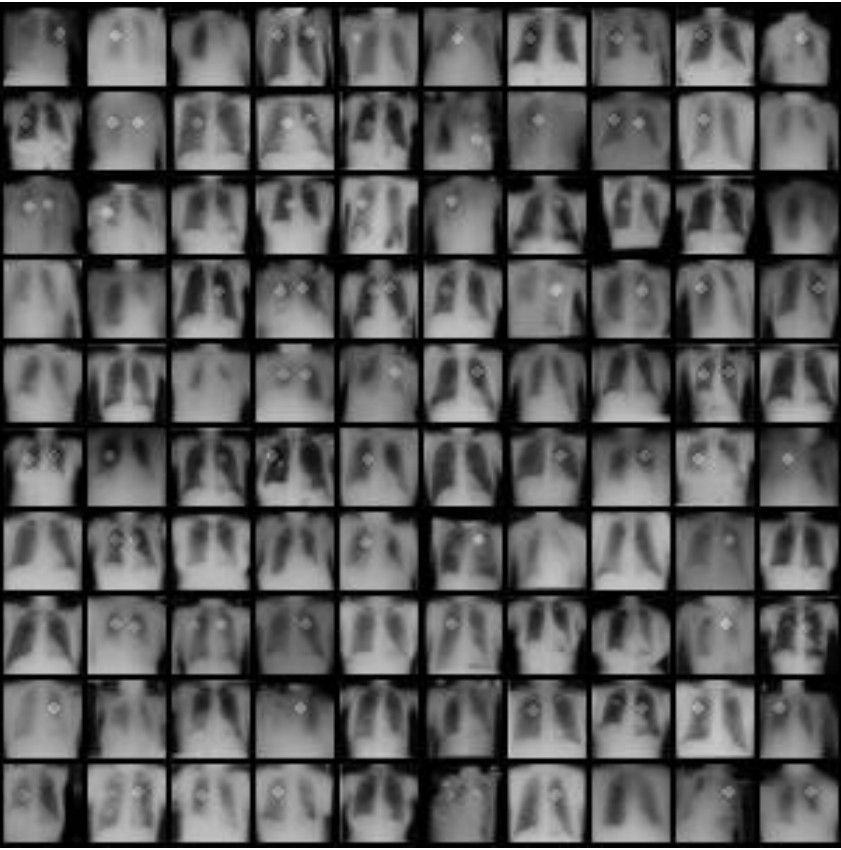

Figure 13: Samples from FFJORD

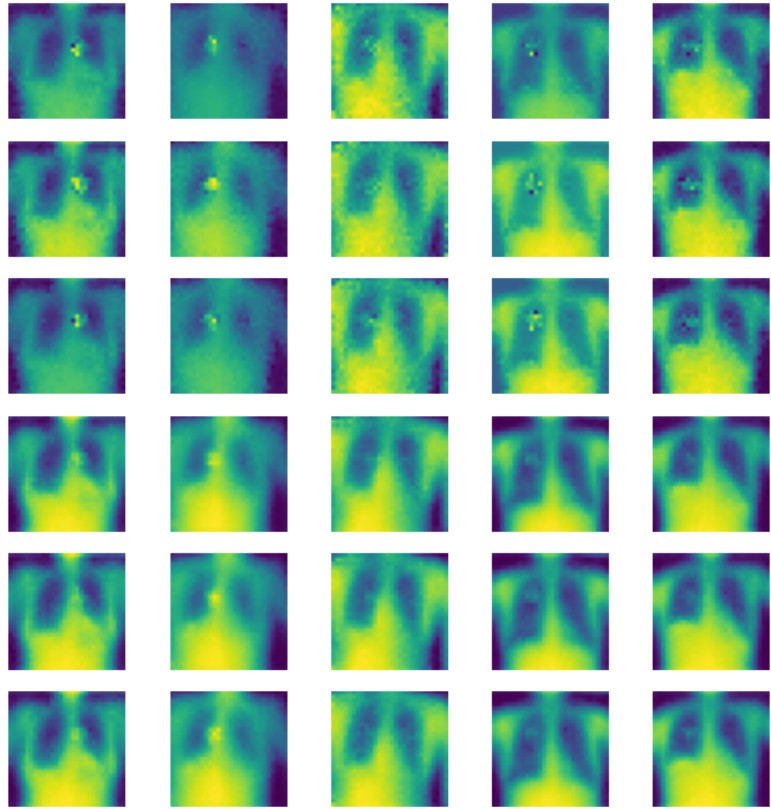

Figure 14: More Causal X-ray samples from the encoder. Top 3 rows are sampled from an encoder that is trained separately. Bottom 3 rows are sampled from a finetuned encoder. More samples further reinforce the conclusion from Figure 6b

.