# OpenReview forum: "High Dimensional Causal Inference with Variational Backdoor Adjustment"
_ICLR.cc/2024/Conference — ICLR 2024 Conference Withdrawn Submission_

### Official Review · Reviewer_m1yM · 2023-10-27

**Soundness:** 3 good
**Presentation:** 2 fair
**Contribution:** 1 poor
**Rating:** 3
**Confidence:** 3

**Summary:**

This paper estimates causal effects in the backdoor setting with high dimensional confounders and treatments. The authors propose using variational inference to efficiently estimate the causal effect with better approximation. The proposed approach considers the confounder as an observed variable. It also includes the optimization of three distributional components named encoder, decoder, and prior that are trained separately using observational data. Finally, the encoder is fine-tuned to get a better estimate of the causal effect/interventional distribution.

**Strengths:**

The derivations of this paper seem technically correct and intuitive. The authors discuss the issues of directly applying the backdoor adjustment formula and joint training. This should help the reader to understand the importance of this problem. The authors provided almost every detail about the experiments which will definitely help reproducing the results easily. The experiment section of this paper is quite rigorous. It contains one synthetic and two semi-synthetic experiments with high dimensional variables. The experiments help readers to understand the algorithm performance for high-dimensional data.

**Weaknesses:**

Here, I provide the weaknesses of this paper and some concerns.

[Section 2]
* The authors should discuss existing neural causal models [1][2][3] that can sample from identifiable interventional distributions. These methods are suitable for training on high-dimensional data and can sample from interventional distributions.

[Section 3.2]
* The authors mentioned, “sampled high dimensional Z will almost never give high probability to p(y | x, z) for a chosen Y and X.” I would request the authors to explain how this leads to a high variance issue.
* It should be more explicitly mentioned that the authors are using a causal sufficient system i.e., all variables [X, Z, Y] are observed.

[Section 3.3]
* The authors mentioned, “the joint training in equation 10 is not guaranteed to converge”. The authors should explain the reasons behind this in more detail.

[Section 4.2]
* Figure 4 a,b, and c should be explained in more detail. It is a little unclear what figures 4 a, and b imply.
* The image qualities of the MNIST and X-ray experiment might be improved by using better neural architectures. If the motivation of this paper includes applying causality in vision with high dimensional images, the proposed method should produce the same quality images as the papers that claim to do intervention without utilizing causal approaches.

Major concerns:
* The authors utilize the backdoor adjustment to estimate the P(Y|do(X)) interventional likelihood. They propose their method for causal graphs that satisfy the backdoor criterion and consider all variables are high dimensional. To my knowledge there exist different neural causal model-based approaches that solve more generic version of this problem.  For example, for a causal sufficient system, i.e., when all variables are observed (same as the authors’ assumption), [1,2] can sample from P(Y|do(X)) for any causal graph. This includes the backdoor graph discussed in this paper as well. For example: The MNIST experiment results should be able to be reproduced with [1,2]. Moreover, [3] relaxes the causal sufficient assumption and can sample from any identifiable P(Y|do(X)) even if there exist unobserved variables in the causal graph. For example, in the front door causal graph: X->Z->Y, X<-U->Y, U is unobserved. [3] can sample from P(Y|do(X)) even if all X, Z, and Y are high-dimensional. These existing works make the novelty of this paper questionable. I would request the authors to distinguish the novelty of this paper compared to the mentioned works.



[1] Murat Kocaoglu, et al. Causalgan: Learning causal implicit generative models with adversarial training.

[2] Nick Pawlowski, et al. Deep structural causal models for tractable counterfactual inference.

[3] Kevin Xia et al The causal-neural connection: Expressiveness, learnability, and inference.

**Questions:**

[Section 3.2]
* How does equation 2,3,4 work for more complex graphs? For example, when there exist multiple variables in the backdoor adjustment set and multiple mediators between X and Y.

[Section 3.3]
* The authors suggested that the trained likelihood-based model components can be plugged as suggested by Equation 4. Can the authors discuss briefly how these components can be combined?
* Why will the separate optimization not lead to the tightest lower bound?
* The authors mentioned that the joint training from the beginning is not guaranteed to converge. What is the guarantee that performing fine-tuning after separate training will always converge for arbitrary complex SCM? ie., the finetuned encoder will better fit with prior p(z) and p(y|x,z) and will obtain a better lower bound in all cases.
* Can the author explain how the distribution over Z becomes untethered from the data?
* What is the role of fine-tuning in this paper besides improving the image quality?

[Section 4]
* What did the authors mean by “binary MNIST images”?

---

> ### Author Response · Authors · 2023-11-20
>
> Dear Reviewer m1yM,
>
> Thank you for the feedback. Responding point by point:
>
> Weakness:
>
> [Section 2]
>
> Thank you for pointing out these works. We will include a more thorough discussion of neural SCMs.
>
> [Section 3.2]
>
> Monte Carlo with an uninformative prior is not useful because it will require an extremely large number of samples. For a fixed sample size, sampling from the posterior will result in an estimate with reduced variance.
>
> Although we several times state that the variables are observed, we also will include the terminology “causal sufficient system”
>
> [Section 3.3]
>
> The argument can be summarized as follows: under joint training, the parameters of $p(y|x,z)$ and $p(z)$ are set to maximize the lower bound in Equation 4 and are no longer bound to the data distribution. In this setting, there exists a trivial parameterization that estimates $\log p(y|x)$. It is possible for $\log p(y|x) > \log p(y|do(x))$, so then it is possible to converge to the wrong solution.
>
> [Section 4.2]
>
> The main result from the MNIST experiment is Figure 4c. By construction, the MNIST observational data will only have X and Z that are consecutive in number, e.g. ([1, 2], [2, 3], [3, 4], etc), and consequently Y will be composed as such. Figure 4c shows that if we model the observational distribution p(y|x), these consecutive numbers will be more likely, whereas when modeling the interventional distribution $p(y|do(x))$ this is not the case.
>
> Major Concerns:
>
> We acknowledge that works [1, 2, 3] are more general in that they apply to different DAGs. However, we maintain that VBA is different because rather than focus on sampling, we can evaluate interventional likelihood. [1] does not explicitly model the likelihood since it is GAN based. [2] estimates likelihood of each of the components, and allows for sampling from counterfactual SCMs, but it does not model interventional likelihood explicitly. Similarly, [3] is the most general and can estimate $p(y|do(x))$, but it does so by sampling. In our work, we have included sampling as a baseline, but going forward will be more explicit with comparing with existing neural SCMs.
>
> Questions:
>
> * In theory, we can generalize our variational inference to the front door graph, but that is beyond the focus of our work. By “more complex graphs”, we mean that the graph can be more complex but Z must be a sufficient adjustment set.
> * In Objective 9, we have parameters $\phi,\theta,\gamma$. These parameters can represent their own gradient-based architecture that can be trained with backpropagation.
> * To clarify, joint training can converge to the wrong distribution. If out encoder $q(z|x,y)$ is expressive enough, we can show that the fine tuning objective will converge to $p(y|do(x))$ in the limit of infinite data.
> * As previously argued for joint training, $p(y|x,z)$ can be parameterized as $p(y|x)$ leading to an inconsistent estimate
> * As shown in the linear Gaussian setting, finetuning will lead to a better estimate of $p(y|do(x))$
> * Binary MNIST is MNIST with pixels taking on values 0 or 1

---

> > ### Comment · Reviewer_m1yM · 2023-11-21
> > **Follow up on major concerns**
> >
> > I thank the authors for their responses and the efforts they put into their paper.
> >
> > The authors said that their algorithm evaluates interventional likelihood rather than focusing on sampling. To my understanding, for high dimensional variables the final goal is to generate samples. How would the authors utilize the interventional likelihoods? What advantage does the proposed algorithm have compared to the sampling-based algorithms [1,2,3] for higher dimensional data? How does estimating likelihood offer better sampling?
> >
> > "[2] estimates the likelihood of each of the components, and allows for sampling from counterfactual SCMs, but it does not model interventional likelihood explicitly." - evaluating interventions is part of evaluating counterfactuals. If [2] can perform counterfactual sampling they should be able to generate interventional samples as well.
> >
> > -A reviewer

---

> > > ### Author Response · Authors · 2023-11-22
> > >
> > > Interventional likelihood $p(y\mid do(x))$ can be used to model causal effect. In our paper, we have an example where a treatment can be high dimensional, i.e. annotations on an X-ray, and we want to find its causal effect. As we show in our paper, the variational approach allows us to estimate $p(y|do(x))$, and sampling performs poorly due to the curse of dimensionality. Although we do not claim our goal is to generate counterfactual samples, in general a high likelihood corresponds to better samples.
> > >
> > > Our point about modeling interventional likelihood is specifically about estimating p(y|do(x)). It is possible sample from p(y|do(x)) without estimating it explicitly, so these are separate problems.

---

### Official Review · Reviewer_wak4 · 2023-10-31

**Soundness:** 2 fair
**Presentation:** 2 fair
**Contribution:** 2 fair
**Rating:** 3
**Confidence:** 3

**Summary:**

The paper proposes a variational formulation of the interventional likelihood (Eq. 4) under the backdoor adjustment. The variational lower bound in Eq. 4 depends on three conditional distributions, each of which is initially trained separately using an MLE. After that, the proposal distribution (or the encoder $q(.|x, y)$) is further trained (or fine-tuned in the terminology of the paper), to further optimize the lower bound.

**Strengths:**

The authors show an ELBO for the interventional log-likelihood $p(y|do(x))$  under a measured confounder set $Z$. The authors demonstrate a way to optimize the ELBO using a two-stage optimization procedure: in the first stage, the three models are trained separately using the MLE. In the second stage, the proposal distribution's parameters are optimized to increase the ELBO while holding the other two models fixed.
The experiments show that sampling from $q(z|x, y)$ leads to lower variance than sampling from $p(z)$ directly.

**Weaknesses:**

There are weaknesses in both the theoretical contributions as well as the experimental setup.

## Weaknesses in the theoretical results.

### Re ELBO in Eq. 4:

The main concern with the lower bound in Eq. 4 is when equality will hold, i.e., under what conditions is the ELBO equal to the interventional distribution in Eq. 2? In the subsequent paragraph, the authors say that "penalty incurred by encoder will be its KL divergence with ... $p(z)$".
The is incorrect: the (Jensen) gap between Eq. 2 and Eq. 4 is *not* equal to KL(q(.|x, y) || p(z)). Did the authors mean something else when they use the word "penalty" here?

The gap is different from the standard variational Bayes formulation (where z is latent) where it is KL between q(z) and p(z|X). In this backdoor case, as the authors say, since $Z$ is observed, the evidence (or Eq. 2) itself depends on Z. So it is not clear to me what $q$ would have to be if equality holds in Eq. 4. Can equality ever hold to begin with?

As a related point, is the estimator obtained by solving Eq. 9 even consistent? Assuming that you had access to infinite data and assuming you could optimize the objective perfectly, will the estimator converge to the true interventional distribution (this should only be possible if the equality can hold in Eq. 4)?


### Re Proposition 1:
Prop. 1 is trying to say how jointly optimizing the objective fails. In the proof, the authors use the fact that if $E[p(y|x)] > E[p(y|do(x))]$, the model converges to p(y|x). I do not understand what $E[p(y|x)] > E[p(y|do(x))]$ has to do with the model converging to $p(y|x)$. The model should converge to whatever maximizes the ELBO (which may or may not be p(y|x)). So even if $E[p(y|x)] > E[p(y|do(x))]$, isn't it possible that the maximum of the objective is obtained somewhere else?


### How do you estimate the ATE?:

In causal inference, we usually care about estimating the ATE or conditional ATE (CATE) using the backdoor adjustment. How exactly do you estimate E[Y|do(x)] from the ELBO in Eq. 4? I can see how, for a given value of $x, y$, you can compute the ELBO in Eq. 4, but how do you compute the expectation E[Y|do(x)]?


### Re motivation:

The authors motivate this work by saying that in case of high-dimensional Z and X, directly sampling Z's leads to high-variance. However, for maximizing the ELBO, the authors need to train models of p(z) and p(z|x, y), both of which are difficult tasks in high-dimensional settings.
One drawback is that difficulties in estimating p(z) for high-dimensional Z will itself lead to difficulties in optimizing the ELBO (and the errors should propagate)? This seems to not fix the issue of high-dimensional confounders. Can the authors comment on this?

By contrast, the standard doubly-robust AIPW estimators require modeling E[Y|z, x] and P(X|Z). In case of high-dimensional $Z$ and low-dimensional $X$ (e.g., binary treatments), why would the variational approach be better? It is not even clear (and neither do the experiments provide evidence for this) that the variational approach suggested by the authors works better than just estimating the ATE using \hat{E}[Y|z, x], i.e., just training a model to predict Y from (x, z).

## Weaknesses in the experimental results.


The main concern with the experimental results in the lack of any baselines. The approaches should at least be compared to IPW, AIPW, and standard regression estimators (like modeling E[Y|x, z] using a ML predictor).

The applies to the MNIST and chest X-ray experiments as well. For the MNIST, if you directly trained a model to predict the concatened output image Y from (z, x), does the variational model do better than that? Without such baselines, it is difficult to understand how well your proposed model is doing.

Moreover, other works, e.g., Shi et. al., have also used neural networks for estimating interventional distributions in the backdoor case. The authors should also compare to that.

**Questions:**

I have addressed the questions in the weaknesses section.

---

> ### Author Response · Authors · 2023-11-20
>
> Dear Reviewer wak4,
>
> We appreciate your feedback and have found it very constructive.
>
> Regarding the Jensen gap, and what value $q(z|x,y)$ must take on to achieve equality in the ELBO, when $q(z|x,y) = p(z| do(x), y)$ the lower bound is tight. We will include this result in the paper and attach a short proof in the appendix. Given that the separate training phase yields a consistent estimate of $p(z)$ and $p(y|x,z)$, we will have consistency assuming $q(z|x,y)$ is parameterized by a family of distributions that includes $p(z|do(x),y)$. This is not notably different from the assumption made in VAEs, that $q(z|x)$ and parameterize the true posterior $p(z|x)$.
>
> For the proof that joint optimization is inconsistent, you correctly say that it will not necessarily converge to $\log p(y|x)$. However, the point of the proof is to show the existence of a situation in which it is possible. Thus, by finding such a counterexample, we show that joint optimization is inconsistent.
>
> As for the motivation, you are correct that if you cannot model $p(z)$ and $p(y|x,z)$ well, then VBA will not yield a good estimate. We also acknowledge that our approach was not intended to be used in the typical treatment effect setting in which X is binary and Z is typically a sparse feature vector in which $p(z)$ is difficult to model. We intended VBA to apply in situations for example where Z is an image. In practice, due to the interdependence of pixels, deep generative models are effective in estimating likelihood estimates of images.
>
> On a similar note, ATE is a concept from binary treatment effect estimation. In our method, it would be estimated with $p(Y=1 | do(X=1) - p(Y=1 | do(X=0)$. However, we do not claim that VBA is designed for binary treatment effect estimation, in which there are many approaches as you mention. The concept of ATE is not particularly well defined or useful for high dimensional X, Y, Z.
>
> Although we understand the concern about lack of baselines, it is difficult to find other methods that are applicable to the same problem: estimate $p(y|do(x))$ for high dimensional X, Y, Z. To the best of our knowledge, methods such as “IPW, AIPW, and standard regression estimators” are not applicable to our setting, but we are open to suggestions.

---

> > ### Comment · Reviewer_wak4 · 2023-11-22
> > **Response to reviewers**
> >
> > I thank the authors for their response. I think the experimental baselines are still poor. For this reason, I would not recommend acceptance at this time.
> >
> > Re tightness of the ELBO:
> > Thanks. Just looking at the bound, it is still not obvious why q(z|do(x), y) makes equality hold. Please include a proof in the next iteration of the paper. I agree with you that, in this sense, the ELBO is similar to a traditional VAE.
> >
> >
> > Re "our approach was not intended to be used in the typical treatment effect setting in which X is binary ... for example where Z is an image":
> > I don't quite understand this argument. I think the authors should test their approach even in settings where Z is high-dimension (like an image) and X is binary. In this case, many existing approaches are applicable. For the standard AIPW estimator, you can train models for P(X|Z) and E[Y|X, Z] and just plug them in the AIPW estimator to get the ATE. For high-dimensional Z, you can use ML estimators. This would still make for a good baseline to at least get a sense of how the variational approach does relative to existing methods. If it is not competitive in the binary X case, it might also be less likely to scale to high-dimensional treatments.
> >
> > Re high-dimensional treatment:
> > Even in this case, why can't you train an ML model for the regression E[Y|Z, X] since Z is a sufficient backdoor adjustment set. Then, in your image experiments, you could intervene and just feed the new image X as input. For example, in Fig. 3, why can't you train a generative model to learn Y from X, Z? It seems like with enough data, this model of P(Y|Z, X) should also work.

---

### Official Review · Reviewer_3NGK · 2023-11-01

**Soundness:** 3 good
**Presentation:** 3 good
**Contribution:** 2 fair
**Rating:** 5
**Confidence:** 3

**Summary:**

This paper addresses the challenge of understanding causal relationships in high-dimensional datasets. The paper introduces "Variational Backdoor Adjustment (VBA)," a framework consists of three distributional components: encoder, decoder, and prior. While the terminology may resemble that of Variational Autoencoders (VAEs), the underlying assumptions are different. These components are optimized separately to ensure correct backdoor adjustment over the observed distributions. The paper provides empirical evidence of the effectiveness of VBA in computing backdoor adjustments. It uses synthetic and semi-synthetic datasets, including high-dimensional linear Gaussian data and image datasets, demonstrating improved interventional density estimation through their optimization technique.

**Strengths:**

- The introduction of Variational Backdoor Adjustment (VBA) presents an innovative solution for handling high-dimensional datasets, offering a fresh perspective on causal inference.

- The paper provides empirical evidence of the effectiveness of VBA in computing backdoor adjustments, including synthetic and real-world data scenarios. This empirical support strengthens the paper's credibility.

- The paper appears well-structured and clear in presenting the methodology and its empirical validation, making it accessible to a broad audience.

**Weaknesses:**

1/ There is no comparison with any baseline making it hard to justify for the perforance of the proposed method. I understand that the proposed method is for high dimensional data. But it should be comparable with some baselines. Is it possible to compare with some popular methods such as BART, X-learner, R-learner, CFRnet, TARnet, etc?


2/ It is unclear on how do you calculate causal effect after training the model. Do you use Eq. (4) as an approximation for log p(y | do(x))?

**Questions:**

Please see section Weaknesses

---

> ### Author Response · Authors · 2023-11-20
>
> Dear Reviewer 3NGK,
>
> Your feedback is much appreciated. For point 1, we do not make comparisons to those methods because VBA is designed for high dimensional X, Y, Z. Those methods have inductive assumptions in the binary treatment regime that we do not make, so a comparison is not completely fair. However, seeing as it is a common point of concern, we will include methods that claim to solve the same problem as baselines and are open to suggestions. Just to clarify, for point 2, you are correct that we use Equation 4 as an approximation for $\log p(y|do(x))$.